

# Description and spatial inference of soil drainage using matrix soil colours in the Lower Hunter Valley, New South Wales, Australia

Brendan P. Malone, Alex B. McBratney and Budiman Minasny

Sydney Institute of Agriculture, The University of Sydney, Eveleigh, NSW, Australia

## ABSTRACT

Soil colour is often used as a general purpose indicator of internal soil drainage. In this study we developed a necessarily simple model of soil drainage which combines the tacit knowledge of the soil surveyor with observed matrix soil colour descriptions. From built up knowledge of the soils in our Lower Hunter Valley, New South Wales study area, the sequence of well-draining → imperfectly draining → poorly draining soils generally follows the colour sequence of red → brown → yellow → grey → black soil matrix colours. For each soil profile, soil drainage is estimated somewhere on a continuous index of between 5 (very well drained) and 1 (very poorly drained) based on the proximity or similarity to reference soil colours of the soil drainage colour sequence. The estimation of drainage index at each profile incorporates the whole-profile descriptions of soil colour where necessary, and is weighted such that observation of soil colour at depth and/or dominantly observed horizons are given more preference than observations near the soil surface. The soil drainage index, by definition disregards surficial soil horizons and consolidated and semi-consolidated parent materials. With the view to understanding the spatial distribution of soil drainage we digitally mapped the index across our study area. Spatial inference of the drainage index was made using Cubist regression tree model combined with residual kriging. Environmental covariates for deterministic inference were principally terrain variables derived from a digital elevation model. Pearson's correlation coefficients indicated the variables most strongly correlated with soil drainage were topographic wetness index (−0.34), mid-slope position (−0.29), multi-resolution valley bottom flatness index (−0.29) and vertical distance to channel network (VDCN) (0.26). From the regression tree modelling, two linear models of soil drainage were derived. The partitioning of models was based upon threshold criteria of VDCN. Validation of the regression kriging model using a withheld dataset resulted in a root mean square error of 0.90 soil drainage index units. Concordance between observations and predictions was 0.49. Given the scale of mapping, and inherent subjectivity of soil colour description, these results are acceptable. Furthermore, the spatial distribution of soil drainage predicted in our study area is attuned with our mental model developed over successive field surveys. Our approach, while exclusively calibrated for the conditions observed in our study area, can be generalised once the unique soil colour and

Corresponding author
Brendan P. Malone,
brendan.malone@sydney.edu.au

soil drainage relationship is expertly defined for an area or region in question. With such rules established, the quantitative components of the method would remain unchanged.

## INTRODUCTION

Soil colour is arguably one of the most obvious and easily observed soil morphological characteristics. Soil scientists use soil colour to differentiate genetic soil horizons as well as for the classification of soil types, e.g. *Isbell (1996)*. From a trained or untrained eye, some inference on soils may be made from observation of soil colour in relation to organic carbon content (*Schulze et al., 1993*; *Aitkenhead et al., 2013*; *Pretorius, Van Huyssteen & Brown, 2017*), mineral composition (*Schwertmann & Taylor, 1977*), soil water content and moisture regime (*Bouma, 1983*; *Blavet, Mathe & Leprun, 2000*). Our interest in this study is making inference of a soils' capacity to drain or soil drainage, based on observed characteristics of soil colour.

For agricultural and environmental applications, soil drainage is an important property that affects plant growth, water flow and solute transport in soils (*Kravchenko et al., 2002*). It has long been established that soil colour patterns can be related to a soils' capacity to drain water (*Evans & Franzmeier, 1988*; *Pickering & Veneman, 1984*; *Vepraskas & Wilding, 1983*). Naturally there are exceptions to this, but often, soil colour can be interpreted as a reflection of oxidative and reductive soil processes. Reductive processes are caused by periodic or continuous water saturation. This could be due to position in the landscape (*Chaplot, Walter & Curmi, 2000*) and/or the presence of a permanently or fluctuating water table near the soil surface. Described in *Bouma (1983)*, reductive soil conditions occur when a soil is saturated. Microbial activity depletes the soil of any free oxygen ($O_2$) causing the soil to become anaerobic. Under anaerobic conditions, and in the presence of organic carbon, ferric iron ($Fe^{3+}$) is microbially converted to ferrous ($Fe^{2+}$). This process is referred to as iron reduction and causes the Fe pigmented coatings on soil particles ($Fe^{3+}$ oxides) to dissolve off the particles and into the soil solution. This results in a washed out and ultimately, grey matrix soil colour, indicating the natural colour of the soil mineral grains. In addition, other redoximorphic features such as mottling and precipitation of manganese are symptomatic of soils which experience periodic or prolonged periods of soil saturation.

Explanations for the causes of soil to remain saturated for prolonged periods include the proximity of a water table or a watercourse line. Related to these physical features is the topographical position of a particular site or landscape. For example, soil saturation occurs in the landscape (from *Chaplot et al., 2004*), when the accumulated water flux, the product of the catchment area $A_s$ and the area drainage flux $q$, passing across an element of contour length $b$, exceeds the product of local soil transmissivity $T$ and the local surface gradient $S$ (*O'Loughlin, 1986*). Thus terrain attributes such as

slope gradient, elevation from, and distance to watercourse lines, and terrain wetness index are generally useful for understanding, but more importantly describing the spatial variation of saturated soils in a particular landscape. The other important variable, which determines soil drainage, is related to its permeability (transmissivity of water). Soil texture and pore size distribution are principal factors which determine the ability of soils to transmit water (*Bouma, 1983*).

Soil drainage classes have been used widely in soil survey to characterise the wetness (drainage capacity) of soil and describe the fluctuations and proximity of the water table at site locations (*Kidd et al., 2014*). The distinctions between different drainage classes are based on tacit knowledge of the soil surveyor, or better, through physical measurements. These measurements may include observations of soil water tables via a well or core, and/or measurement of the soil water status. Measuring the soil water status for estimations of drainage class requires prolonged monitoring, and though possible, the procedure is complicated, costly, and time consuming (*Bouma, 1983*). With these logistical issues, it is not surprising that soil colour and assessment of soil redoximorphic features are often used as an indicator for making assessments of soil drainage.

The implementation of quantitative indexes of soil drainage, inferred from soil colour and/or redoximorphic features is not a new concept. Some include that of *Evans & Franzmeier (1988)* which requires numeric indexing of Munsell notation, in addition to information regarding mottle characteristics and abundance. *Blavet et al. (2002)* also numericalised Munsell colour notations in addition to using of a soil redness index (*Torrent et al., 1983*) to derive a continuous index for describing the duration of water-logging. *Chaplot, Walter & Curmi (2000)* developed a continuous index (0–100) of soil hydromorphy based on the cumulative thickness of soil horizons with redoximorphic features, combined with information regarding the Munsell Hue and Value numbers. These studies exemplify the value of using low-cost soil morphological information for making inference of soil drainage characteristics.

Our interest in this study is to develop a different type of continuous index of soil drainage. It is necessarily simple, because the soil database we are using is limited in terms of direct measurements of soil drainage and is inconsistent, even unreliable in terms of descriptions of the abundance or even presence of redoximorphic features such as mottles. In the simplest terms, the drainage index we develop in this study combines some tacit knowledge with actual observations made in the field of the soil matrix colour (each genetic soil horizon), to derive a continuous whole-profile index of soil drainage.

The motivation for deriving a soil drainage index is that we are particularly interested in understanding its spatial distribution across the landscape, as this is probably more useful from a land management and assessment perspective. Studies such as *Schaetzl et al. (2009)* demonstrate this. Furthermore, after a number of years surveying the area described in this study, we have developed a mental concept of how soil drainage varies across the landscape. It is a useful exercise to validate such mental models with empirical information. Given the relationship between topography and soil saturation, there is considerable benefit in applying digital soil mapping methods (*McBratney, Mendonca-Santos & Minasny, 2003*) for inferring the spatial distribution

of soil drainage. A number of studies have constructed soil spatial inference models of soil drainage class using topographical variables (*Kravchenko et al., 2002*; *Campling, Gobin & Feyen, 2002*). *Bell, Cunningham & Havens (1992)* used multivariate discriminant analysis using topography and geological information to spatially predict drainage classes. *Chaplot et al. (2004)* were interested in mapping the soil hydromorphic index using topographic indices derived from the land surface and saprolite upper boundary. Less invasive techniques of mapping soil drainage classes through the use of remote sensing platforms have also been demonstrated (*Peng et al., 2003*; *Cialella et al., 1997*).

The aims of this study are threefold: (1) To develop an index of soil drainage combining tacit knowledge and empirical information of soil matrix colour; (2) to determine whether an empirical relationship exists between the estimated drainage index and landscape features; and (3) to develop a soil spatial prediction function for estimating the spatial distribution of soil drainage across our study area in the Lower Hunter Valley, NSW (New South Wales, Australia).

## MATERIALS AND METHODS

### Study area

The area of this study is the Hunter Wine Country Private Irrigation District (HWCPID), situated in the Lower Hunter Valley, NSW. The HWCPID covers approximately 220 km$^2$ and encompasses the localities of Pokolbin and Rothbury, NSW (32.83°S 151.35°E), which are approximately 140 km north of Sydney, NSW (Fig. 1). Topographically, this area consists mostly of undulating hills that ascend to low mountains to the south–west. The underlying geology of the HWCPID is predominantly Early Permian, with some Middle and Late Permian formations. Described in the Newcastle Coalfield Regional 1:100,000 Geology Map (*Hawley, Glen & Baker, 1995*), the most extensive formation is the Rutherford Formation (Early Permian) which consists of siltstones, marl, and some minor sandstone. Much of the southern and eastern part of the HWCPID is underlain by the Rutherford Formation. Other extensive formations include: the Mulbring Siltstone (Late Permian siltstones), the Branxton Formation (Middle Permian conglomerates, sandstones, and siltstones) and the Farley Formation (Early Permian silty sandstones). These formations occupy the north-western extents of the HWCPID. In terms of land use, dryland agricultural grazing systems are predominant, followed by an expansive viticultural industry. While most of the land has been dedicated for these uses, tracts of remnant natural vegetation (dry forest) are apparent, particularly towards the south-western area—which is bordered by Broken Back Range, Werakata National Park situated to the east, and some areas situated in the northern extents.

Our knowledge of the soils across the HWCPID was first informed from legacy soil survey which is described in detail within the Soil Landscapes of the Singleton 1:250,000 Sheet Map and Report (*Kovac & Lawrie, 1990*). This knowledge has since evolved through annual soil surveying campaigns by students and members of our research group, which began in 2001 and continue to the present time. These annual surveys, while concentrated to the south of the study area, form a densely populated database of soil information and descriptions. This information and soil knowledge has been

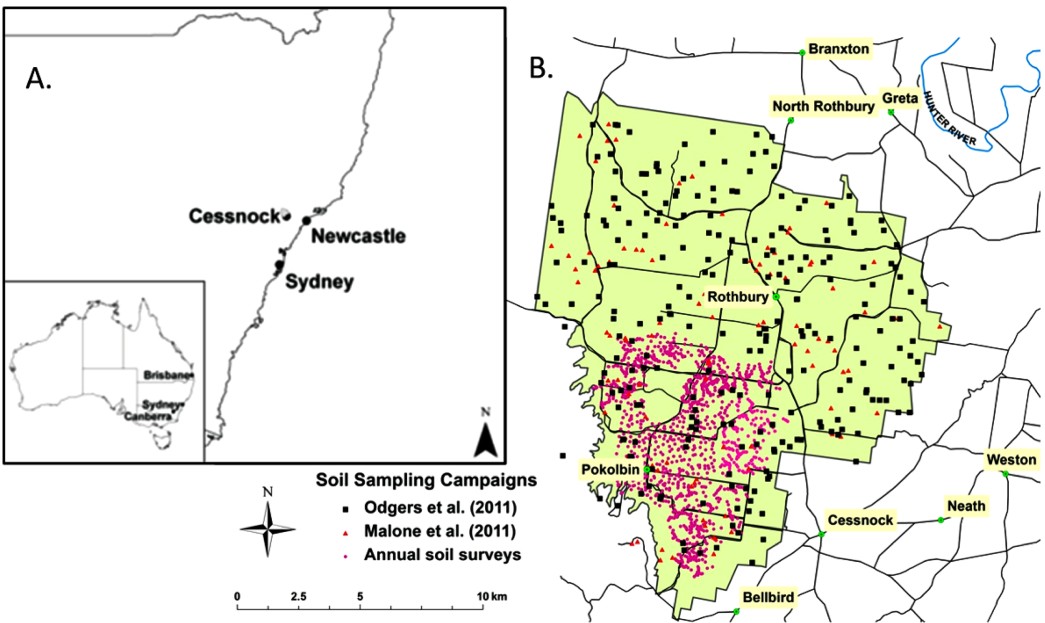

**Figure 1 Locality map.** (A) The Hunter Wine Country Private Irrigation District relative position in Australia and New South Wales. (B) The Hunter Wine Country Private Irrigation District (shaded green) and surrounding localities. Sampling locations of the three survey campaigns: *Malone et al. (2011)* 34 soil profiles, *Odgers, McBratney & Minasny (2011)* 251 soil profiles, and 1,261 soil profiles from annual surveys. Black lines indicate roads. Blue lines are major watercourses.

supplemented with two area-wide soil surveys of the HWCPID which have been described in *Malone et al. (2011)* and *Odgers, McBratney & Minasny (2011)*. Based on these various soil surveying campaigns we have found—based on the sub-order level of the Australian Soil Classification system (*Isbell, 1996*)—that the most dominant soils across the HWCPID are both Brown and Red Dermosols and Chromosols. Generally, Dermosols and Chromosols are the most prolific; there are few Kurosols by comparison. Hydrosols and Rudosols are few, but generally concentrated near watercourse lines. Calcarosols are also few, yet exist in areas where the Rutherford Formation exists, particularly where the occurrence of the calcareous marl parent material is present. Corresponding WRB (*Food and Agriculture Organisation (FAO), 1998*) soil classes to the ASC soil orders are: Calcisols (Calcarosols), Luvisols (Chromosols and some Dermosols), Acrisols (Kurosols and some Dermosols), Fluvisols (Hydrosols), and Regosols (Rudosols).

## Conceptual model of the spatial distribution of soil drainage

In the HWCPID, we have often observed a common sequence of soils down hillslopes, which indicate varying degrees of soil drainage. Morphologically, this sequence can be observed as changes in the matrix soil colour. For example, red coloured soils are observed a lot on hilltops and crests. Brown and yellow soils can be found further down the hillslope, and often grey and black soils are found on the foot slopes near watercourse lines. This sequence of soil colour is not uncommon down a hillslope in other parts of the world (e.g. *Simonson & Boersma, 1972*; *Bouma, 1983*; *Kravchenko et al., 2002*).

**Table 1 Site and soil morphological information for three soil profiles down a catena in the HWCPID.**

| Site: Crest | Projection: WGS84 | Latitude: −32.7903°S | Longitude: 151.3190°W |
|---|---|---|---|
| **Elevation:** 104 m.a.s.l | **TWI:** 16 | **MRVBF:** 0.00 | **VDCN:** 31 m |
| **Soil type:** Red Dermosol | | | |
| **Horizon** | **Depth (cm)** | **Clay %** | **Munsell notations matrix (moist)** |
| A1 | 0–3 | 31 | 10YR 2/2 |
| B21 | 3–63 | 38 | 5YR 3/4 |
| B22 | 63–79 | 37 | 7.5YR 5/4 |
| B23 | 79–103 | 40 | 7.5YR 4/3 |
| **Site:** Mid-Slope | **Projection:** WGS84 | Latitude: −32.7930°S | Longitude: 151.3201°W |
| **Elevation:** 81 m.a.s.l | **TWI:** 18 | **MRVBF:** 0.50 | **VDCN:** 9 m |
| **Soil type:** Brown Dermosol | | | |
| **Horizon** | **Depth (cm)** | **Clay %** | **Munsell notations matrix (moist)** |
| A1 | 0–14 | 29 | 10YR 3/2 |
| B21 | 14–23 | 35 | 10YR 4/4 |
| B22 | 23–51 | 37 | 10YR 4/6 |
| **Site:** Flat | **Projection:** WGS84 | Latitude: −32.7969°S | Longitude: 151.3208°W |
| **Elevation:** 70 m.a.s.l | **TWI:** 20 | **MRVBF:** 2.92 | **VDCN:** 0.07 m |
| **Soil type:** Grey Dermosol | | | |
| **Horizon** | **Depth (cm)** | **Clay %** | **Munsell notations matrix (moist)** |
| A1 | 0–11 | 25 | 10YR 3/4 |
| B21 | 11–42 | 39 | 10YR 4/4 |
| B22 | 42–77 | 32 | 10YR 4/2 |

Note:
TWI, topographic wetness index; MRVBF, multi-resolution valley bottom flatness Index; VDCN, vertical distance to channel network.

In Table 1, site and soil morphological data are provided for three soils, developed from the same parent material (siltstone), at different positions of a hillslope in the HWCPID. These data are not in isolation; rather they represent a common occurrence, not of soil type, but soil colour change and presumably soil drainage. On the crest is a Red Dermosol, which then grades into a Brown Dermosol at the mid-slope position (MSP), followed by a Grey Dermosol on the flat, near a watercourse line. Terrain variables: topographic wetness index (TWI), multi-resolution valley bottom flatness index (MRVBF), and vertical distance to channel network (VDCN) highlight some topographical information which may provide further explanatory evidence for describing this sequence of soils and associated soil drainage. For example, TWI and MRVBF, both indices for describing the movement and concentration of water in the landscape, increase down the hillslope, i.e. soils in the mid-to-low parts of the hillslope accumulate and concentrate more water than soils on or near the hillcrests.
Similar to a soil drainage index, the soil hydromorphic index by *Chaplot, Walter & Curmi (2000)* requires information regarding redoximorphic features, i.e. mottling for its derivation. Because we cannot rely on this data in our database (as such features have not been consistently nor accurately recorded), we need to derive another index, based exclusively on the soil matrix colour. Further in the discussion we propose an approach how to incorporate such features within our simple index. Our drainage index ranges continuously between and including the values of 5 and 1. The conceptual model of soil water drainage in the HWCPID and exemplified with the data in Table 1, is that 'red' soils have the highest drainage index value of 5, 'brown' soils (4), 'yellow' soils (3), 'grey' soils (2), and 'black' soils (1). This index implies that 'red' soils drain better than 'brown', which drain better than 'yellow' soils and so on. 'Black' soils are the poorest in terms of soil drainage because it is these soils that appear to be saturated permanently and as a consequence have accumulated carbon. The soil drainage index has been designed for where descriptions have been made for each genetic soil horizon of a soil profile (but it may also be applied where soil is observed as regular or at specific depth intervals). Derivation of the soil drainage index is now described in the following methodological sections.

## Derivation of the drainage index

### The data

In this study we use soil data collected from three major soil surveying campaigns conducted in the HWCPID. In total, these campaigns have amounted to 1,546 individual soil profile observations and descriptions (Fig. 1). The breakup of these profiles is: 34 come from the work by *Malone et al. (2011)*; 251 from the work of *Odgers, McBratney & Minasny (2011)*; and 1,261 from annual soil survey work for the years between 2001 and 2011. For each of these soil profiles, data was recorded for each genetic horizon. Our primary interest is in the matrix soil colour of each horizon, particularly the moist colour, which was recorded on the basis of matching the observed soil colour with a colour chip on a Munsell HVC (Hue, Value, Chroma) colour chart. We disregarded horizon descriptions where the lower boundary did not exceed 40 cm from the top of the soil profile. We also disregarded horizon descriptions of semi- and unconsolidated parent materials which in Australian soil nomenclature are described as B/C and C horizons, respectively (*The National Committee on Soil and Terrain, 2009*). For example, if the first horizon of a particular soil profile was 0–55 cm, then it would be included in the drainage index model. If a soil profile had a sequence of horizons measuring: 0–30, 30–75, 75–120, and >120 cm (which was found to be bedrock), then the drainage index would only consider the observed data from 30 to 120 cm. After this filtering process, we ended up with 3,731 soil horizon data with moist soil colour descriptions to work with.

Munsell HVC soil colour descriptions are not conducive for quantitative studies. Therefore, we performed a conversion from the Munsell HVC colour space to the CIELAB colour space (*Robertson, 1977*; *International Commission on Illumination (CIE), 1978*). The CIELAB colour space can describe any uniform colour space by the three variables:

$L^*$, $a^*$, and $b^*$. Each variable represents the lightness of the colour ($L^* = 0$ yields black and $L^* = 100$ indicates diffuse white), its position between red/magenta and green ($a^*$, negative values indicate green while positive values indicate magenta) and its position between yellow and blue ($b^*$, negative values indicate blue and positive values indicate yellow). The non-linear equations for converting from Munsell HVC to CIELAB are described in *Viscarra Rossel et al. (2006)*. First Munsell HVC are converted to the CIE XYZ colour space based on a fitted neural network model of known XYZ values and corresponding Munsell soil colour chips, which are derived from the Munsell Conversion program Version 6.41 (http://www.gretagmacbeth.com). Standard CIE (1978) equations are then used to transform from CIE XYZ to CIELAB. Because a model based approach (neural networks)—rather than a physical relationship or direct correspondence—are used to transform from Munsell HVC to CIE XYZ, the prediction will inevitably be uncertain to some degree. The extent of this uncertainty is not known. *Viscarra Rossel et al. (2006)* do state however that the conversion was adequate.

One of the problems with descriptions of soil colour is that they are subjective and can be ambiguous estimates. Each individual's perception of colour is different, which will result, for the same soil, often quite different predictions of soil colour. In order to work with this type of data, our drainage index is rooted in fuzzy set theory (*Zadeh, 1965*), meaning that some of the ambiguity and uncertainty in soil colour prediction can be dealt with by allocating each observation, membership to multiple defined classes. Therefore the first step in defining a drainage index entails designating centroids or archetypal soils for each soil colour/drainage class. Using the unconverted data (i.e. the Munsell HVC colours), we designated each observation to a particular colour class based on the colour groupings of *Northcote (1979)*. From summary statistics, we came up with three centroids (the three most frequently observed) for each colour class to represent the reference or archetypal soil colours (Table 2). Three reference colours (15 in total) for each colour class was a pragmatic decision based on the fact that we wanted to derive an appropriate configuration of centroids within the $L^*$, $a^*$, and $b^*$ feature space.

With the reference colours established, we then estimated the Mahalanobis distance of each observation to each reference colour:

$$d_i = \sqrt{(\boldsymbol{x_i} - \boldsymbol{c_j})^T \boldsymbol{S^{-1}} (\boldsymbol{x_i} - \boldsymbol{c_j})}$$
$$i = 1, \ldots, N; j = 1, \ldots, C$$

(1)

where $d$ is the Mahalanobis distance between the multivariate vector $\boldsymbol{x}$ (here an observed $L^*$, $a^*$, and $b^*$ vector) and reference colour vector $\boldsymbol{c}$ ($L^*$, $a^*$, and $b^*$). $\boldsymbol{S}$ is the variance–covariance matrix of $N$ observed $\boldsymbol{x}$. The result here is an $N \times C$ matrix ($\boldsymbol{D}$) where each element $d_{ij}$ represents the Mahalanobis distance of each observed horizon colour $i$ to each reference colour $j$.

The measure of similarity (or membership) of each horizon observation to each reference colour is estimated as:

**Table 2 Reference soil colours.**

| Reference colour | Munsell® colour | | | CIELAB | | |
|---|---|---|---|---|---|---|
| | Hue | Value | Chroma | $L^*$ | $a^*$ | $b^*$ |
| Red 1 | 5YR | 4 | 6 | 41.55 | 16.37 | 29.43 |
| Red 2 | 2.5YR | 4 | 8 | 42.23 | 20.39 | 34.10 |
| Red 3 | 2.5YR | 4 | 6 | 41.63 | 19.36 | 24.63 |
| Brown 1 | 7.5YR | 4 | 6 | 41.41 | 13.84 | 33.59 |
| Brown 2 | 10YR | 4 | 6 | 41.23 | 11.75 | 37.06 |
| Brown 3 | 10YR | 5 | 6 | 51.45 | 10.65 | 36.50 |
| Yellow 1 | 10YR | 6 | 8 | 62.06 | 11.45 | 49.99 |
| Yellow 2 | 10YR | 6 | 4 | 61.50 | 8.34 | 22.99 |
| Yellow 3 | 10YR | 6 | 6 | 61.65 | 10.41 | 36.50 |
| Grey 1 | 10YR | 4 | 2 | 41.03 | 6.91 | 9.15 |
| Grey 2 | 7.5YR | 4 | 2 | 41.16 | 8.31 | 8.33 |
| Grey 3 | 10YR | 6 | 2 | 61.67 | 4.84 | 9.73 |
| Black 1 | 10YR | 2 | 2 | 19.07 | 15.73 | 8.13 |
| Black 2 | 7.5YR | 3 | 2 | 30.70 | 10.91 | 7.82 |
| Black 3 | 10YR | 3 | 2 | 30.56 | 9.69 | 8.58 |

**Note:**
Reference soil colours for each soil colour class in both Munsell HVC and CIELAB colour space notation. Note that the reference colours are grouped following the colour groupings that were created by *Northcote (1979)*.

$$u_{i,j} = \frac{1}{1 + \sum_{l=1, l \neq j}^{C} \left( \frac{d_{ij}}{d_{il}} \right)^{\frac{1}{m-1}}}$$

$$i = 1, \ldots, N; j = 1, \ldots, C; l = 1, \ldots, C-1$$

(2)

where $u_{i,j}$ is the similarity of horizon observation $i$ to reference colour $j$, and where $d_{il}$ is the Mahalanobis distance of $i$ to the other reference colours $d_{il}$. Thus observations close to (as determined by the Mahalanobis distance) a reference colour will have a higher similarity than those observations more distant. The fuzzy exponent $m$ determines the level of similarity fuzziness of $i$ to each reference colour. A value of 1 for $m$ will result in all $u_{i,j}$ converging to either 0 or 1, which implies a crisp partitioning of the observations to the reference colours. Conversely, $m$ values approaching infinity will create similarities with complete overlap such that an observation will have equal similarity to all reference colours. In this study, we pragmatically set $m$ to 1.5 on the basis that we did not desire to crisply partition the observations, yet still allow for some overlap to the reference colours.

## Estimation of drainage index for each horizon and subsequently each soil profile

The drainage index ranges continuously between and including the values of 5 and 1. As per the conceptual model of soil water drainage in the HWCPID, *red* soils have the highest drainage index value of 5, *brown* soils (4), *yellow* soils (3), *grey* soils (2), and *black*

soils (1). For each horizon the drainage index is calculated as a weighted average based on the degree of similarity to each reference colour. Such that:

$$DI_i = \sum_{i=1}^{N} u_{ij} RC_j \qquad (3)$$

where $DI$ is the drainage index and $RC$ refers to the reference soil colour $j$. While not considered in this study, the presence of mottles could potentially be included in this index with the following equation:

$$DI_{i(rx)} = DI_i \times Rp \qquad (4)$$

Here $DI_{i(rx)}$ is the drainage index incorporating information about the proportion of mottles within the soil matrix (expressed as a percentage), and $R_p$ is simply 1—the observed proportion. Of course this equation would need to be tested against real data.

Continuing on from Eq. (3), because we need to derive a whole profile drainage index value we need to aggregate each $DI$ calculated at each horizon for each soil profile $P$. However, we want to preferentially weight the observed $DI$ values such that observations at depth are given more weight to those higher up the soil profile. Firstly, for each horizon in soil profile $P$, a vector based on the observed upper and lower horizon boundaries is created and then summed. For example, in $P$, a particular horizon is observed to occur from 45 to 75 cm. The summed vector of this sequence (i.e. 45, ..., 75) is 1,860. We may denote this summed vector as $SV$. Therefore the whole-soil profile drainage index value can be calculated as:

$$DI_P = \sum \frac{SV_h}{\sum\limits_{h=1}^{Z} SV_h} \cdot DI_h$$
$$h = 1, \ldots, Z$$

where $DI_P$ and $DI_h$ are the drainage index value/s for the whole-profile and genetic soil horizons respectively of a soil profile $P$.

## Correlation of the drainage index with environmental variable

The ultimate aim of this paper is to derive a drainage index map for the HWCPID. As a preliminary step we wanted to investigate the relationship (using Pearson's coefficient of correlation) of the derived drainage index with a suite of environmental covariate information. In this study, this covariate information is exclusively derived from a digital elevation model (25 m ground resolution) sourced from the NSW Government. Informed from similar work of mapping soil drainage and hydromorphy (such as *Kravchenko et al., 2002*; *Campling, Gobin & Feyen, 2002*; *Chaplot et al., 2004*), from the digital elevation model we derived a number of potentially useful primary and secondary terrain variables: Elevation (E), slope gradient (S), slope length (SL), slope height (SH), MSP, terrain wetness index (TWI), VDCN, MRVBF, analytical hillshading (AH). These indices were derived using the terrain analysis modules of SAGA GIS (http://www.saga-gis.org/), and described in more detail in Table 3.

**Table 3 Description of topographic variables.**

| Topographical variable | Unit | Description |
|---|---|---|
| Elevation (E) | Metre | Metres above sea level; derived from a digital elevation model |
| Slope gradient (S) | Degree | Measured in degrees, is the first derivative of elevation in the direction of greatest slope |
| Mid-slope position (MSP) | Dimensionless | Is commonly considered in topoclimatic analysis, to cover the warmer zones of slopes. This parameter assigns mid-slope positions with 0, whereas maximum vertical distances to the mid-slope in either valley or crest directions are assigned with 1 in order to represent the temperature drop towards upper and lower parts of a slope |
| Slope height (SH) | Metre | A relative elevation variable which is estimated from calculating the vertical distance from the base of a slope to the crest of the slope |
| Slope length (SL) | Metre | A measure of the distance from the point of origin of overland flow to either of the following, whichever is limiting for the major part of the area under consideration: (a) the point where the slope decreases to the extent that deposition begins, or (b) the point where runoff enters a well-defined channel that may be part of a drainage network or a constructed channel such as a terrace or diversion (*Wischmeier & Smith, 1978*) |
| Terrain wetness index (TWI) | Dimensionless | A secondary landform parameter which uses catchment area and slope gradient which estimates for each pixel, its tendency to accumulate water |
| Vertical distance to channel network (VDCN) | Metre | Difference between elevation and an interpolation of a channel network base level elevation. Knowledge of the spatial distribution of channel networks (lines) is therefore necessary for this parameter |
| Multi-resolution valley bottom flatness (MRVBF) | Dimensionless | Multi-resolution valley bottom flatness is derived using slope and elevation to classify valley bottoms as flat, low areas (*Gallant & Dowling, 2003*). This is accomplished through a series of neighbourhood operations at progressively coarser resolutions with the goal of identifying both small and large valleys. MRVBF has been used extensively for the delineation and grading of valley floor units corresponding to areas of alluvial and colluvial deposits. High values of MRVBF indicate relatively low, flat areas of the landscape |
| Analytical hillshading (AH) | Dimensionless | Analytical hillshading derived from the DEM is used as a surrogate for positional aspect and considered in topoclimatic analysis. The parameter is calculated by positioning a light source at given azimuth (measured in degree clockwise from the north direction) and elevation (measured in degree above the horizon). Subsequently by setting azimuth to 315 and elevation to 45, we can determine where in the landscape north facing slopes are positioned, which generally receive the most sunlight |

**Note:**
Description of topographic variables used in this study.

## Mapping the drainage index

We use a digital soil mapping (*McBratney, Mendonca-Santos & Minasny, 2003*) framework for the spatial interpolation of the drainage index across the HWCPID to the same resolution as the topographic variables (25 m). The dataset of 1,546 profiles was randomly split into calibration (70%) and validation (30%) datasets. For calibration, the soil spatial prediction function employed here was a regression kriging model. Using the covariates described above, we used Cubist models to identify any deterministic relationship with the drainage index at each of the observed soil profiles. Cubist is a prediction-oriented regression model that is based mostly on work by *Quinlan (1992)*. Although it initially creates a tree structure, it collapses each path through the tree into a rule. A regression model is fitted for each rule, based on the data subset defined by the rules. The set of rules

are pruned or possibly combined, and the candidate variables for the linear regression models are the predictors that were used in the parts of the rule that were pruned away. The residuals from the Cubist model were investigated for spatial autocorrelation as a means to detect any additional (random) spatial trend of the drainage index not detected from the covariates. We used geostatistics and locally fitted variograms (based on the exponential model) for spatial interpolation (kriging) of the residuals across the entire HWCPID. The sum of the outputs from the deterministic modelling and residual kriging resulted in a final drainage index map.

The validation dataset was withheld from the calibration procedure. Using measures such as the root mean square error (RMSE) and concordance coefficient we compared the regression kriging predictions at each of the validation profiles with their 'observed' value. The RMSE measures the differences between predicted and observed values and is estimated by:

$$\text{RMSE} = \sqrt{\frac{\sum_{i=1}^{n} \left( z_{pi}(s) - z_i(s) \right)^2}{n}}$$

where $z_{pi}(s)$ and $z_i(s)$ are the predicted and observed values of validation point $i$ and $n$ is the number of validation points. The concordance coefficient measures the fidelity of the observations and the predictions to a 1:1 line (*Lin, 1989*).

The implementation of methods in this study (where previously not already stated) were carried out using the R statistical software (*R Core Team, 2015*) for general statistical analyses and mapping. The R package 'Cubist' (*Kuhn et al., 2016*) was used for fitting the Cubist model. VESPER geostatistical software (*Minasny, McBratney & Whelan, 2005*) was used for the local fitting of variograms and kriging.

## RESULTS

Pearson's coefficient of correlation between the derived soil profile drainage index and each of the covariate data sources from highest to lowest were: TWI (−0.34), MSP (−0.29), MRVBF (−0.29), VDCN (0.26), SH (0.22), SL (−0.18), S (0.11), E (0.09), and AH (−0.03). These correlation coefficients indicate some general features of soil drainage in the HWCPID, for example there is a positive correlation of the drainage index with vertical proximity to watercourse lines. Indices such as TWI and MRVBF, which inform us about the hydrological characteristics of the area, are negatively correlated with the drainage index. Thus based on landscape position, where the soil is more prevalent to concentration of water, the drainage index is also lower. The correlations of the slope indices S, SH, and MSP with the drainage index indicate a relationship whereby gentle slopes (relatively low S and SH, and high MSP) soils more likely to have a lower drainage index. Similarly, longer slopes (SL) result in a negative correlation with the drainage index.

Fitting of the Cubist model to the calibration data resulted in the partitioning of two simple rules for the spatial distribution of the drainage index. Each rule defining a different regression model:

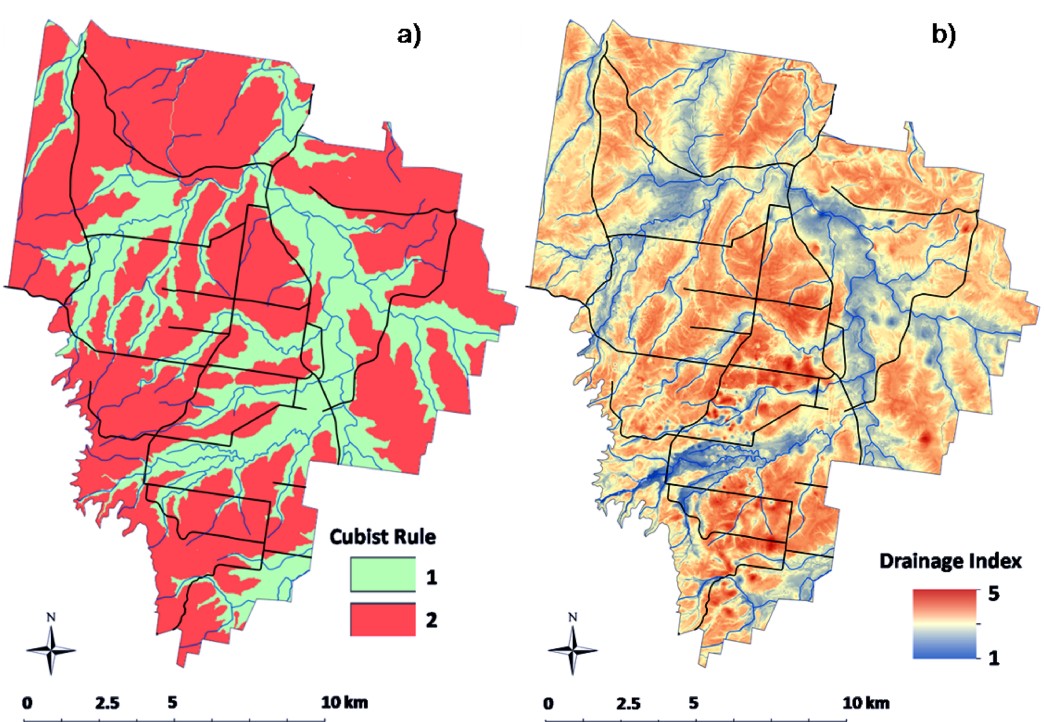

**Figure 2 Drainage index map.** Spatial map of the application of Cubist rules across the HWCPID (A). Map of the soil drainage index across the HWCPID (B). Black lines indicate roads. Blue lines indicate watercourse lines.

**Rule 1**

*if VDCN ≤ 7.8 m*

*then*

$DI = 6.58 + 0.06(VDCN) - 0.20(TWI) - 0.02(E) - 0.80(AH)$

**Rule 2**

*if VDCN ≤ 7.8 m*

*then*

$DI = 6.58 - 0.11(TWI) - 0.01(E) - 0.09(MSP) + 0.001(VDCN)$

These simple linear regressions are pre-empted by a recursive split of all data based on a threshold value of 7.8 for the VDCN. Essentially this means that vertical proximity to a watercourse line is a defining characteristic of soil drainage. Common parameters to each linear model were VDCN, TWI, and E, while AH was only included in the first rule and MSP was only included in the second rule.

Examining where each Cubist rule was applied shows clearly the relationship of the rules with proximity to watercourse lines (Fig. 2A). For approximately one-third of the area, rule 1 was applied.

The associated drainage index map which resulted from the regression kriging model is shown in Fig. 2B. With the blue lines indicating the watercourse lines, it is clear from the map that proximity to them has a considerable effect on the soil drainage. From a basic statistical analysis, where rule 1 was applied, the mean drainage index was 2.70

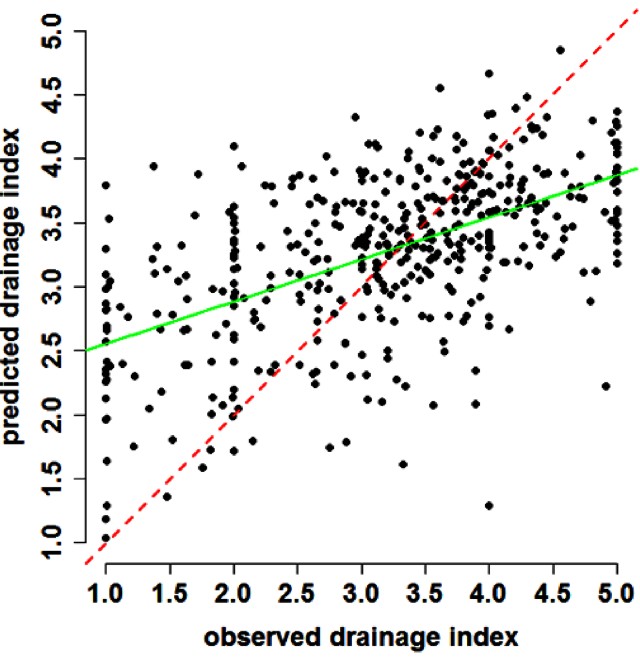

**Figure 3 Scatterplot of results.** Observed and fitted plot of drainage index based on regression kriging predictions for the validation dataset. Red dashed line indicates a line of concordance (1:1 relationship). Green solid line is a regression line for the linear relationship between the observed and predicted drainage index.                              

with 95% of the area between 1.85 and 3.60. Where rule 2 was applied, the mean drainage index was 3.40 with 95% of the area between 2.51 and 4.00.

Validation of the regression kriging model based on 446 withheld data indicated a RMSE of 0.90, meaning that, the predictions of the drainage index on average deviate approximately 0.9 away from the observed value. The concordance between the observed and predicted values was a reasonable 0.49. The plot in Fig. 3 show the observations and corresponding predictions with respect to the 1:1 relationship (draw as a red dashed line). Predictions appear to be strongest around drainage index values between 2.5 and 4.0. From visual inspection of the plot it is clear there does not seem to be any bias, such as prevalence for over or under predictions (mean error was calculated as −0.08). A linear model fitted to the observed and fitted data resulted in a coefficient of determination of 31%, indicating a reasonable correlative relationship (green solid line).

## DISCUSSION

The continuous index of soil drainage proposed in this study requires little information other than tacit knowledge of soil drainage spatial variability, and observed soil matrix colour descriptions. The fundamental limitation of this is that we have assumed there is a direct correlation between soil colour and soil drainage. There is good physical evidence though to support this relationship (e.g. *Bouma, 1983*; *Evans & Franzmeier, 1988*). It is likely, since we have been able to establish a correlative relationship between our index of soil drainage and some topographical variables, that some validation in terms
of soil water measurements or measurements of water table proximity is warranted in further studies.

There is significant value from the land management perspective in quantifying soil drainage as a spatially continuous variable across the landscape. In this study, we have avoided mapping the well-known and established drainage classes. While there may be value in doing this, the grading between classes is qualitative and in the absence of direct measurement, allocation to a particular class is a subjective designation. Nevertheless, it was by necessity (due to the data used) that we had to develop our own model, which by default treats soil drainage as a continuous variable. Subsequently, mapping the continuously varying drainage index across the HWCPID revealed spatial patterns more attuned to what one would observe in the landscape; that is, continuously varying rather than discreetly apportioned. In the HWCPID, it is believed that discreet variations of soil drainage are the exception rather than the rule.

On the basis of using digital soil mapping methods, we have been able to validate quantitatively the spatial model of soil drainage. Comparatively with other digital soil mapping studies, the results found in this study are acceptable (*Grunwald, 2009*). Other studies that have examined the relationship between soil drainage and environmental information have reported stronger correlations than that reported in this study (*Campling, Gobin & Feyen, 2002*; *Chaplot et al., 2004*). It is suspected that scale may be one cause for this discrepancy. For example, we have attempted to describe the variations of soil drainage over a much larger area of land which we know to be rather complex (topographically *and* lithologically). While these results may be improved upon, the map, from a soil surveyor's perspective, adequately coincides with the knowledge we have developed over the years of survey in the HWCPID.

In terms of the spatial prediction model, we used the Cubist models as an attempt to mirror what a soil surveyor would observe in the landscape. That is, given particular combinations of features or characteristics of the landscape, a particular soil or characteristic of the soil will behave similarly. The quantitative interpretation of this and what was found in this study, was that vertical distance to a channel network was a divisive and important physical attribute determining the estimation of soil drainage; such that, given a certain threshold, different predictive models were applied. More generally, we have found that using such rule-based spatial prediction functions makes them more interpretable (from the soil survey perspective) and particularly useful for digital soil mapping.

Correcting the deviation between what was observed (from the data) and what was predicted using a spatial model is a worthwhile pursuit. What was clear in this study is that the observed variations of soil colour described something much more complex than what the spatial model was able to describe. There could be many reasons and explanations for this. One of them is that soil colour alone and attribution thereof can have significant influence on the interpretation of soil processes. While fuzzy set theory is embedded within our model, which by definition embraces the subjectivity around soil colour attribution, our model is by no means immune to poorly attributed soil colour descriptions. Ultimately, this can have flow-on effects when soil colour is then used in some sort of quantitative model, e.g. soil drainage (*Chaplot et al., 2004*). *O'Donnell et al.*

*(2010)* have proposed a standardised procedure of soil colour attribution based on image processing. Or perhaps usage of a colorimeter would be enough to make standardised assessments of soil colour. Currently, standardised assessments of soil colour are not made during soil survey around the world, so it is likely it will be some time before we can test the applicability of our method using such assessments.

The spatial model of soil drainage in this study principally used topographical variables as predictive covariate information. We also incorporated a regression kriging model with the intention of further modelling spatial trend that was not detectable from the topographic information. Regression kriging made some improvement of the prediction in comparison to just using a deterministic model. Nevertheless, due to a limitation in the availability of additional sources of predictive information, we were unable to explore more complex relationships of soil drainage with other environmental variables. For example, parent material or underlying geology has been shown to be a useful variable (*Bell, Cunningham & Havens, 1992*). Intuitively, different lithologies will impart differing soil physical characteristics, such that the drainage characteristics of a soil developed from limestone will be different from those developed on siltstone or sandstone etc. The best available geological survey of the HWCPID (1:100,000; *Hawley, Glen & Baker, 1995*) informs us that while siltstones are the most predominant lithology, there are also sandstones and silty sandstone parent materials. A limitation of our mental model of soil drainage is that it has been refined where the lithology is predominantly siltstone—as most of the soil sampling has been conducted on this lithology. There is potential bias regarding estimation of soil drainage to contend with where other lithologies are found. However, the question is whether the current geological survey could be used to refine our model of soil drainage? It is unlikely that it would, because while instructive, it is neither comprehensive or of the appropriate scale. Furthermore, soil processes such as colluviation and alluviation have often created soil profiles of complex and mixed lithology that is near impossible to disentangle from geological survey maps. We envisage that in the future, gamma-radiometric survey will provide us with information regarding the lithology and lithological processes at the necessary detail to be included within our soil drainage model. Gamma radiometry refers to the measurement of naturally occurring gamma radiation which is emitted from the ground surface (*Cook et al., 1996*). Such information has been shown to describe the distribution of soil-forming materials and weathering processes over large areas (*Wilford, 2012*).

In the absence of detailed lithological information, a pragmatic solution may be to examine whether the digital mapping of soil texture grades (or soil variables derived from them such as bulk density etc.) are useful for interpreting variations of soil drainage. In the HWCPID, where soil textures are recorded predominantly as that derived from hand bolusing, we need to explore methods of how to incorporate these data within a digital soil mapping framework.

## CONCLUSION

By necessity of the data available, we have developed an index of soil drainage which incorporates tacit knowledge of the soil surveyor and observed soil matrix colour. Soil

drainage is evaluated as a whole-profile, weighted combination of the soil colour at each generic soil horizon. Fuzzy set theory is built into the drainage index model as a means to dampen the subjectivity of soil colour attribution. We believe the approach can be generalised to other areas once the unique soil colour and soil drainage relationships have been defined by an expert.

In our study, we found that the topographical variables most strongly correlated with soil drainage are TWI, MSP, MRVBF, and vertical distance above channel network. Cubist models were used to model the relationship of the drainage index with a suite of topographic variables with the dual purpose of understanding the spatial variation of soil drainage and to validate our mental model of soil drainage developed over the years from successive field surveys. Validation of the spatial model of soil drainage was adequate in consideration of the scale of mapping and nature of the data. The associated map corresponds meaningfully to what we have generally observed in the field. The incorporation of new information specifically from gamma-radiometry or soil texture may be useful solutions in improving our understanding of soil drainage in the HWCPID.

### Funding
The authors received no funding for this work.

### Competing Interests
Budiman Minansy is an Academic Editor for PeerJ.

### Author Contributions
- Brendan P. Malone conceived and designed the experiments, performed the experiments, analysed the data, contributed reagents/materials/analysis tools, prepared figures and/or tables, authored or reviewed drafts of the paper, approved the final draft.
- Alex B. McBratney conceived and designed the experiments, contributed reagents/materials/analysis tools, authored or reviewed drafts of the paper, approved the final draft.
- Budiman Minasny conceived and designed the experiments, contributed reagents/materials/analysis tools, authored or reviewed drafts of the paper, approved the final draft.

### Data Availability
Bitbucket data repository
https://bitbucket.org/brendo1001/hv_soilcolour.

### Supplemental Information
Supplemental information for this article can be found online at http://dx.doi.org/10.7717/peerj.4659#supplemental-information.

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
