# Peer review of "Description and spatial inference of soil drainage using matrix soil colours in the Lower Hunter Valley, New South Wales, Australia"

_PeerJ, doi:10.7717/peerj.4659_

## Round 0.1 · original submission · Major Revisions

The manuscript addresses an interesting subject: the relationship between soil colour and drainage conditions. Although this is a well-studied and well-known subject, the attempt to develop a drainage index defined from a commonly and easily determined soil property is noteworthy and useful for soil surveyors. The study handles a huge number of soil profiles and uses a sound methodology.

Taking into account the opinions of three reviewers, I consider the manuscript acceptable for publication after major revisions.

I exhort the authors to take into account the very helpful comments by all 3 reviewers. Consider them as aids to improve the manuscript. However, take them as recommendations, not as mandatory: you are the authors and the ultimate responsible of the manuscript.

One of my main questions, which I would like to be addressed in the article, is to which extent the results could be extrapolated to other areas (different climate, different lithology, …).

Agreeing with reviewer 1 that the use of a colorimeter would provide more accurate measurements of soil colour, it would be difficult to make such a high number of colour measurements. Munsell soil colour is a commonly available soil property and makes the index useful for soil surveyors. The recommendations of the first reviewer may be very useful for further study.

One thing I disagree with one of the reviewers is that the “Materials and methods” section is a chaos. I find it quite helpful, particularly for a reader not familiarised with some statistical methods or with soil mapping.

I agree with all three reviewers that, besides the Australian system, soil taxonomic names according to international systems, such as US Soil Taxonomy or WRB, should be given. Moreover the topographic parameters (TWI, MRVBF, MSP, AH, etc.) must be defined.

The transformation of Munsell colour to CIELAB parameters is a crucial part of the article. More information on the method of transformation should be given and the possible error in this transformation addressed. As reviewer 1 states, the reference Vissarra Rossel (2006) cannot be reached. Include also information on DEM.

Take into account the recommendations of reviewers 2 and 3 regarding bibliography and Figures.

The bibliography is somewhat outdated. It should be updated. Only 6 out of 37 bibliographical references are more recent than 2010. I am surprised that you do not cite your own paper (Kidd et al., 2014) in Soil Research. Moreover I can make a couple of suggestions:
Pretorius, M.L., Van Huyssteen, C.W. & Brown, L.R. Environ Monit Assess (2017) 189: 556. https://doi.org/10.1007/s10661-017-6249-z
Aitkenhead, M.J., Coull, M., Towers, W., Hudson, G., Black, H.I.J. 2013. Prediction of soil characteristics and colour using data from the National Soils Inventory of Scotland. Geoderma 200–201, 99-107. https://doi.org/10.1016/j.geoderma.2013.02.013

I agree with reviewer 2 that the relationship between predicted and observed drainage index appears to be weak. Figure 3 shows that the deviation of predicted values from observed values is positive for low DI and negative for high DI. Most predicted values lie between 2 and 4, while the observed values range between 1 and 5. This must be highlighted (and, if possible, explained) in the text. Moreover, some predicted values (in the lower right part of the Figure) are abnormally low; this must also be highlighted.

I agree with reviewer 2 that the manuscript is well-written and easy to understand. Notwithstanding, the English language needs some revision. I exhort the English-speaking authors to thoroughly revise the manuscript. Besides the remarks made by reviewers 2 and 3, I have a few more remarks regarding the language:
- In line 81, “principle” should be “principal”
- In line 124, “,” after threefold should be “:”
- In line 127, for consistency, “Develop” should be “To develop”
- In line 165, “there” should be “there are”
- In line 167, “is” should be deleted (“is exists” is redundant)
- In line 177, “is” should be “are”
- In line 219, “than” should be “then”
- In line 342, “a positive correlation with the drainage index with vertical proximity”, should be “a positive correlation of the drainage index with vertical proximity”
- In line 347 “to” should be deleted
- In line 368 “affect” should be “effect”

Please give the meaning of acronyms the first time you use it. For example, NSW: it is not obvious for a non-Australian reader.

I agree with reviewer 2 that the conclusions are well stated, although you could write them in a continuous way, without points (it is up to you).

Reviewer 1 ·

Basic reporting

English is correct and the literature references are correct. However I have found several references that does not fit well with the meaning that the author explained. I cannot find the correlation between the authors' idea and the idea reflected in the literature. In addition, I have encouraged the authors to use an international soil classification instead of Australian or add more information (analytical data about their soils) because it cannot be enough for me in order to identify the characteristics of soils.
The structure of article is perfect and the figures and tables should be improved because there is not sufficient important information for corroborating the outcomes.
In reference to the general paper. They define a future lines that must be applied to the research in order to obtain a final complete paper. There is not numerical data in the paper and the information is based on the tacit knowledge of the researchers that have been different according to the database. In addition, this database is not well explained in the paper

Experimental design

In my opinion the article should be planned by the use of a colorimeter in the experience. As authors explained, the tacit knowledge is the basis of this study, and in my opinion this is a problem because we do not have any information in order to corroborate that tacit knowledge. In addition the authors established a series of statements that cannot be extrapolated to other experiences because they have defined the relationship between soil colour and drainage classes according their experiences. Not always red soils have good drainage and black soils show bad drainage to apply the 1 to 5 index.
In addition, there is not a good description of soils, there is not a definition of why they have decided to classify the soils in those groups and the topographic parameters are not explained. In addition, we do not know if data show a normal behaviour. Which is the level of statistically significance? They have tested the kriging method, and cartography is well developed because they show outcomes of RSME and the correlation between predicted and observed. However it is a part of the article.
I have not known which has been the survey method. How many people have collaborated in the survey? And, there are students that have participated in the study but we do not know which their level about soil colour identification or classification is. In addition material and method section should be reordered because it is a chaos in order to know what the authors have done.
As authors explain in the article there are some limitations that they recognize. In that case, I encourage them to recover the experience and try to afford that limitations in order to create a new article that could be more complete.

Validity of the findings

In this case, the findings cannot be reproduced according their experience in other areas, in addition they did not explain which are the database or information that can make readers to adapt this experience to other areas. I have mentioned that in my opinion data is not robust because there is a lack of information to corroborate the outcomes. The conclusion section should be rewritten and the in reference to statistically, I do not know if data are normal, and which is the significance level of the information that the authors give the readers. There is not any table of statistics summary and the data about colour it is not well defined or explained.

Additional comments

In general, the article tries to establish a direct relationship between colour and drainage in a specific area of Australia. The main problem to classify the article as REJECT has been that there is not enough information to corroborate the outcomes. As authors mention, they include a tacit knowledge of the soil surveyor and observed soil matrix colour. The problem is that we cannot know which the tacit knowledge is. There is not any information about it. In the same way, the authors explain in the discussion that the model shows a limitation due to their mental model of soil drainage. In addition, they hope in the future to add soil texture or other gamma-radiometry data. I recommend that they try to add more information as mineral and physical properties of soils, and in that situation prepare a complete article. I cannot understand that why BLACK soils have to be “badly drained” because this statement it is only used in their tacit Knowledge. I know a lot of BLACK soils that shows a good drainage and they are black due to the wet conditions (without permanent saturation conditions). The landscape position is one of the soil forming factors that affect soil formation but there is more other factors that can favour a good/bad drainage. Soil particle composition, presence of impermeable layers, etc. At the end the authors stablished the relationship “BAD DRAINAGE=SATURATION”. Why do you decide that correlation soil colour – drainage? Red=5, brown=4, yellow=3, grey=2 and black=1. There is not any data that corroborate that relation.
You should add information about soil profiles. Texture, porosity… but there is not anything to corroborate that yours classification (1-5) is correct.
There are some concepts that they try to relate in order to develop a complete experience but they are only correct in their tacit knowledge and in their local conditions. There is not true that red soils ALWAYS show a good drainage. I have known soils in Mediterranean area that shows red colours but they shows a bad drainage due to the presence of impermeable layers in the profile. The climatic conditions do not allow the development of anaerobic conditions and in this case the statement that you are using for soil colour relation with drainage cannot be applied. In my opinion the simple relationship between colours and drainage capacity cannot be extrapolated to other situations where the natural conditions are different. In the same way, if I want to reproduce the experience, which is the tacit knowledge that we have to add? Which are the profile descriptions?
Another thing that it is strange for me, is the colour measurement. The article is based on colour and the authors decided to identify Munsell colour instead of using a colorimeter which give the CIELAB coordinates. You wrote in the article that “MUNSELL soil colour descriptions are not conductive for quantitative studios”, but you planned an experience in which you need numerical data. Which is the mistake that can be produced in the change of coordinates ( MUNSELL-CIELAB)? There is not mentioned in the article. You have add a reference to the Method of Viscarra Rossel, but the link it does not work (and it does not the “,”). I have looked for information in an article of Viscarra and this is the reference that he added: Viscarra Rossel, R. A. (2004), ColoSol, CSIRO Land and Water, Canberra. However, I do not know the mistake or problems derived in the automatic process of calculation. Why did you not use a colorimeter?
The authors could have avoided the subjective vision of soil colour by the humans with the use of colorimeter. Have you tested the correct assignation of colour by the students in the 2001-2011 campaigns? There is an article published by the American Journal of Soil Science where the outcomes about soil colour reveal that student only fit well 26% of soil samples (in the correct chip). It is clear that trained researchers can suffer the same situation and they do not fit well the soil colour, but in this case the identification mistake is lower and it is demonstrated in several articles. Had you trained your students before the campaigns? If not, we have to assumed that can appear two “blanks” in your experience, one for the wrong soil colour chip identification and other by the differences in the adjustments between MUNSELL and CIELAB. And all of this situations can be avoided.
In addition there are several things in the text that it will be some confusion. In my opinion there are some parts of the text that are conclusions or advises that you introduce along the different sections.
“Furthermore, after a number of years surveying the area described in this study, we have developed a mental concept of how soil drainage varies across the landscape. It is a useful exercise to validate such mental models with empirical information.”
“Further in the discussion we propose an approach how to incorporate such features within our simple index”
“Because we cannot rely on this data in our database” Which are the information in your database?

You have used the Australian classification but if an article is addressed for an international audience you should try to classify in an INTERNATIONAL SOIL CLASSIFICATION, or in the case that you want to preserve the Australian classification (that can be correct) you should give us some information of physical and chemical properties in order to know which type of soils you present. By this reason, the soils description shows a lack of information. In the same way, you say that there are some horizons that are disregarded. How many horizons did you disregard? At the end I do not know which the total amount of SOIL profiles/horizons is (used for your study).

This is a list of several things that in my opinion should be improved or clarified. There are more and I have only explain some of my doubts.

Abstract
Line 21  “NSW”. What does it mean? It appears in several parts of the paper and I cannot know its meaning.
Line 26-27  Can you explain me why did you give more weight to the master horizon or depth horizon against upper horizon? Which was the reason? Was it depending the position on the landscape? I asked you all of this questions, because in the text there is no data about if this “assumption” is correct or not. I have to trust your words, because there is not any evidence that explain your decision. For example, could it be an impermeable layer in your profiles? In your area you have among others, calcareous materials that can produce argillic horizons. It depends which type of clay particles you can find impermeable layers. In other way, there is a movement of carbonates that cam produce a petrocalcic horizon. Does it happen in your area?
Line 26  You have written that the estimation drainage index incorporates the whole profile descriptions, but in the paper we can only observe a short description of 3 profiles with the unique data: colour.
Line 50  the reference of Isbell (1996). It is clear that authors are Australian and it is an example. However, as your article is addressed to an international audience, in my opinion it would be better if you change your local classification by a worldwide classification as FAO or USDA. I am sure that you can find some examples and for the audience it will be more understandable.
Line 50  I am not sure that an untrained eye can identify soil texture and mineral composition.
Line 52  I have checked Blavet et al 2000 (https://www.researchgate.net/publication/29636156_Soil_colour_variables_as_simple_indicators_of_the_duration_of_soil_waterlogging_in_a_West_African_catena)
And if it is correct Blavet et al do not relate texture and soil colour as you explain in the text. I can be wrong, so I ask you to correct me if it is this situation. If it is not correct, please CHANGE or rewrite what you want to express. I think that there is not a direct relationship between soil colour and texture. I have studied red soils from sandstones with a sand texture and white soils with sand texture too.
Lines50-53  This sentence should be reformulated for a better understanding:

“From both a trained and untrained eye, some inference about a soils organic carbon content (Schulze et al. 1993), mineral composition (Schwertmann and Taylor 1977), soil water content (Bouma 1983) and soil texture (Blavet et al. 2000) may be made from observation of soil colour.”

An example can be:

“From both a trained and untrained eye/ From a trained or untrained eye, some inference on soils may be made from observation of soil colour in relation to organic carbon content (Schulze et al. 1993), mineral composition (Schwertmann and Taylor 1977), soil water content (Bouma 1983) and soil texture (Blavet et al. 2000), among others characteristics ”

Line 53-54  The sentence “Our interest in this study is making inference of a soils’ capacity to drain or soil drainage, based on observed characteristics of soil colour” should appear at the end of the introduction section. It is not usual that in the state of art the authors include the goal. Therefore, rewrite the paragraph and send it to the end of the section.

Lines 58. As the authors explain, soil colour patterns can be related to soil’s capacity to drain water. In that situation, the authors can explain some examples or add the references as they have put, but in this case it is not necessary to add inside the brackets “for example”. It is well known that in those publications there will be examples. In the same way, can you say that all red soils have a good drainage? (SOILS DERIVED FORM RED SAND MATERIALS – SOILS DERIVED FROM MARLS) In my opinion the sentence “It has long been established that soil colour patterns san be related to a soils’ capacity to drain water” must be nuanced.

Line 59  “This is because soil colour can be interpreted as a reflection of oxidative and reductive soil processes”. With this sentence the authors are

Lines 65-67  Superscripts of Fe

Lines 73-77  can be summarized in one sentence. It is not necessary to add all of that parameters.

Lines 182 -188  This sentences should be add in the 303-308 lines, because you repeat some variables. Which are the units of the variables? The authors do not specify.

Lines 215-223  I cannot understand why you limited the data according to your explanation on lines 215-223. Can you explain why did you take this decision?

Outcomes

The data has a normal behaviour? You have used Pearson’s coefficient but in any situation you have referred to the statistical outcomes. We suppose that each Pearson coefficient has been statistically significant but which is the level of this significance?

Line 353  7.8 (units)? Km? m? cm?


Conclusions


Should be rewritten in a paragraph. Along the text you have added some sentences that should appear in the conclusions (lines 390-391, or 411).

In my opinion TABLE 1 SHOWS A LACK of information. They have more than 1546 profiles and the authors only exposed 3 profiles and only data of horizon, depth and colour. I would like to know with is the number of MUNSELL colour notations in the set of profiles, which is the relationship between colour and landscape position, or between colour and geological materials.
Authors should try to summarize the data (that we do not know that they have) in a table in order to understand something and do not have to believe all that they present without any information to corroborate the paper.

Table 2 is correct, but which was the reason for selecting this 3 groups inside each group of colour? How many samples inside the pull of data are directly correlated with this exact data?

Figure 1. There is not any reference about the location of this area in Australia. There is not any level curve and the scale it is not so good. In my opinion there is not a representative definition of the area. I would know if the samples are at the bottom flat, or in the hillslope, and this is not reflected in the map. In my opinion should be improved in order to try to communicate more information, not only a map with several points that explained the campaigns.

Best regards

Reviewer 2 ·

Basic reporting

a. English language usage: The manuscript is well-written and easy to understand. The authors described their methods and results well. I did note a couple grammatical or typographic errors (e.g., a missing apostrophe in “soils organic carbon content” in line 51 and “attune” should be “attuned” in line 403), but these were minimal and did not impede communication.
b. Intro & background: The introductory section of the manuscript provides a good summary of literature on the relationship between soil color and drainage class, which is useful in understanding the context of this research. The literature referenced is relevant to the points cited. I did note several instances where references are cited in the text, but not included in the list of references; e.g., Torrent et al. 1983 in line 96, Schaetzl et al. 2009 in line 111, Peng et al. 2003 and Cialella et al. 1997 on p. 123. I suggest that the authors carefully ensure that all cited references are included in the list of references (and vice versa).
c. Structure: The structure of the manuscript is a standard structure for research articles. It is clear and easy to follow.
d. Figures: The figures are relevant and the captions are good. I do have the following suggestions to improve figures:
i. Figure 1 could be cartographically improved considerably to make it easier to read and interpret. Varying the weight of some of the lines would help. An inset map or accompanying figure would be useful to an international audience as to where this site is located. Any geographical feature referred to specifically in the text should also be included on the figure (e.g., the Broken Back Range and Werakata National Park in line 148). It is possible that roads are not very important to what this figure needs to communicate; omitting them from the figure would make it less busy and easier to read, though perhaps they are useful for comparing locations with Figure 2. I only see one blue line on the figure (labeled “Hunter River”): is it the only watercourse, or are the watercourses covered up with the roads? It seems to me that the watercourses are much more important than the roads to this study.
ii. Figure 2 is fine, though perhaps omitting the roads would make the figure clearer.
iii. I would prefer to also see maps of the various terrain variables that are included in the model (referred to in lines 304-306). This would help the reader to get a better understanding of the topography of the study area and how the drainage index compares to the terrain variables. If they are not included due to space limitations in the main article, it would be nice to see them at least in the supplementary material, but I would prefer to see them in the body of the article itself.
e. Raw data: The raw data is supplied.

Experimental design

a. The research is original primary research within the scope of the journal.
b. The research question is well defined, relevant, and meaningful. The wording of aim #2 in lines 125-126 is awkward; perhaps it could be re-worded to clarify it. The manuscript states how the research fills an identified research gap.
c. The investigation is performed to a high technical and ethical standard.
d. The methods are well and thoroughly described. I believe that it would be possible to replicate the study from the description provided.

Validity of the findings

a. The data are robust, statistically sound, and controlled.
i. The authors note, correctly, that there are issues with a dataset that relies on visual matching of soils to color chips, especially when done in relatively uncontrolled circumstances, as would be present in a soil survey situation, but I believe that their method does what can be done with such a dataset. It is helpful that there are quite a large number of measurements. On the other hand, while the authors seem satisfied with the results presented in Figure 3, the relationship between predicted and observed drainage index appears to me to be weak at best. It would not encourage me to attempt to use this method to predict drainage using this method. But of course the authors must present the results they obtained.
ii. I did not note any description of the digital elevation model used to obtain the terrain variables (source, spatial resolution, etc.). Since terrain indices can vary considerably with different DEM characteristics, especially spatial resolution, the authors should include this information in the description of their methods.
b. The conclusions are well stated, linked to the original research questions, and limited to supporting results.

Additional comments

a. I thought this manuscript was well-done. With a few minor revisions, I believe it should be published.
b. While the method of producing the drainage index described here is interesting and could prove useful to other researchers, the specific index is likely to be heavily dependent on the particular soil types present in this study area, and would therefore be difficult to apply directly in another situation.

Reviewer 3 ·

Basic reporting

Comments to the manuscript entitled “Description and spatial inference of soil drainage using matrix soil colors in the Lower Hunter Valley, New South Wales, Australia"

In the manuscript, the authors attempted to draw a map of the degree of soil drainage by using matrix soil colors in the Lower Hunter Valley, NSW. The manuscript is well organized with clear objectives and conclusions. However, the manuscript did not bring a better understanding of the empirical knowledge of the relationships between soil color in the study area and the degree of soil drainage. This is mainly due to the lack of the data on the soil water regimes such as volumetric water content at the time of soil sampling, infiltration rate and depth of water table that can be related to other soil attributes. So the map of the soil drainage index presented in Figure 2b appears to be based on the untested assumptions that soil color is mainly originated from the reduction of Fe oxides and accumulation of organic matter, both having been influenced by soil water regimes, and that soil drainage is also influenced by the distance to the nearest river. The soils on the natural embankment near a river are often sandy with a higher relative altitude and lower water table than those in the floodplains. This phenomenon seems to be unaccounted for in Figure 2b. Because some of these inherent drawbacks are pointed out clearly in the first paragraph of the Discussion, the manuscript can be acceptable with some revisions as follows.

Small comments
1) L65 and other parts: ferric or ferrous iron (Fe3+, Fe2+) upper case letters for 3+ or 2+.
2) L125-126: Is “if” necessary?
3) L140: ands some minor sandstone, and?
4) L164-166: Besides the Australian system, soil taxonomic names according to the international systems such as US soil taxonomy should be described.
5) L322: “to for” correct?
6) L380: 0.31?
7) Table 2: According to a Munsell color chart, 5YR 4/6 is referred to as “reddish brown”, although it is classified as Red 1 by the authors. Please check your Munsell color chart, and add the exact color names corresponding to the 15 classes based on the color chart in addition to your classifications.
8) Figure 1: Add the number of soil sampling points in each sampling campaign. A blue line is found only at the upper right position (Hunter River) out of the study area. Is it OK?
9) Figure 2 (caption): Black lines indicate “roads”?

Experimental design

no comment

Validity of the findings

no comment

---

## Round 0.2 · accepted · Accept

The revised manuscript has considerably improved relative to the first version. So I consider the manuscript acceptable for publishing. I have just corrected a few grammatical or typographic errors.